# Explaining and Measuring Social-Ecological Pathways: The Case of Global Changes and Water Security

**Thomas Bolognesi** [1,*], **Andrea K. Gerlak** [2] **and Gregory Giuliani** [3]

1   Institute for Environmental Sciences - GEDT & Department of Political Sciences and International Relations, University of Geneva, 1205 Geneva, Switzerland

2   Udall Center for Studies in Public Policy, University of Arizona, Tucson, AZ 85721, USA; agerlak@email.arizona.ed

3   Institute for Environmental Sciences, EnviroSPACE Lab., University of Geneva, 1205 Geneva, Switzerland; gregory.giuliani@unepgrid.ch

*   Correspondence: thomas.bolognesi@unige.ch; Tel.: +41-223-790-814

**Abstract:** The Social-Ecological Systems framework serves as a valuable framework to explore and understand social and ecological interactions, and pathways in water governance. However, it lacks a robust understanding of change. We argue an analytical and methodological approach to engaging global changes in SES is critical to strengthening the scope and relevance of the SES framework. Relying on SES and resilience thinking, we propose an institutional and cognitive model of change where institutions and natural resources systems co-evolve. Our model of change provides a dynamic understanding of SES that stands on three causal mechanisms: institutional complexity trap, rigidity trap, and learning processes. We illustrate how data cube technology could overcome current limitations and offer reliable avenues for testing hypotheses about the dynamics of Social-Ecological Systems and water security by offering to combine spatial and time data with no major technical requirements for users.

**Keywords:** social-ecological system; water security; governance; institution; learning; data-cube

## 1. Introduction

Pressures on natural resources are increasing and a number of challenges need to be overcome to address growing population in a period of environmental variability. The key to sustainable development is achieving a balance between the exploitation of natural resources for socio-economic development, and conserving ecosystem services that are critical to humans' wellbeing and livelihoods [1]. The policy objective of water security is symptomatic of this challenge [2–4].

Water is a crucial natural resource supporting life on Earth and underpinning equitable, stable, and productive societies and ecosystems [5]. The cumulative effects of overexploitation, pollution, and challenges posed by climate change demand careful monitoring and assessing of trends and evolution of water resources [6]. This is an essential pre-condition to ensure a sustainable use and access to water that can serve as a basis for water security. This important role has been recognized as one of the seventeen Sustainable Development Goals (SDGs) defined by the United Nations. From a territorial management perspective, to support monitoring and reporting commitments and obligations, managers and decision-makers need information that is synoptic, consistent, spatially explicit and sufficiently detailed to capture anthropogenic effects [7].

The Social-Ecological Systems (SES) framework combines different components to organize consistently the analysis of human and nature co-evolution [8]. It is designed as a diagnostic tool to

support identification of relevant variables that can determine sustainability in SESs and allow for analysis of relationships among levels and scales. As such, it contributes to a common framework to facilitate multidisciplinary research efforts, better understanding and study of the complexity of these systems [8], and cross-institutional comparisons and evaluations [9]. However, SES lacks a robust understanding of change because of a limited number of empirical studies and its use as a checklist [10]. We argue an analytical and methodological approach to engaging global changes in SES is critical to strengthening the scope and relevance of the SES framework. Identifying the methodological needs for carrying-out large-N (>100 observations) dynamic analyses of SESs is needed for knowledge accumulation.

In this paper, we put forward a way to explain change and dynamics in SES that could be testable across many cases. Current literature offers mainly in-depth investigations highlighting mechanisms and triggers of changes. However, there is a need to scale-up the knowledge toward more generalizable results [11,12]. Relying on SES and resilience thinking, we propose an institutional and cognitive model of change for SES where institutions and natural resource systems co-evolve. Our SES model of change offers a dynamic understanding of SES, and stands on three causal mechanisms: institutional complexity trap, rigidity trap, and learning processes. We argue that this model of change can enable systematic and explanatory tests, and combine them.

Methodologically, we highlight current methodological limitations regarding measurement heterogeneity and time-limited analysis and emphasize the need for carrying out studies about SES dynamics. Then, we provide insights on how to operationalize this, mainly through measurement rather than methodological tools. Data cube technology could overcome current methodological limitations. This technology is a new big data tool providing ready-for-analysis datasets that combine Earth observation data over years as well as space [13,14]. It requires no major technical skills from users or researchers and offers reliable avenues to test hypothesis about the dynamics of social-ecological systems and water security by offering to combine spatial and time data with no major technical requirements for users. We pose water security as an illustration of this analytical and methodological perspective because it is conducive to analyzing human and nature interdependencies [3,4], thereby highlighting the spatial and time dimensions of SES. Indeed, water availability changes over space, time, and quality in consequence of human and natural trends [5,15–18].

First, we propose an institutional and cognitive model of change to feed the social-ecological system framework and frame analysis of SES evolution. Then, in Section 3, we discuss the current methodological barriers to accumulate knowledge about SES evolutions and share recommendations to overcome them. We present the data cube technology and approach in Section 4 as a promising tool to put into practice our recommendations and test the proposed model of change. Finally, we conclude by identifying the next steps and avenues.

## 2. Social-Ecological Systems and Evolution: Theoretical Framework

In this paper, we propose a robust understanding of change for social-ecological systems analysis. We argue that an analytical and methodological approach to engaging global changes in SES is critical to strengthening the scope and relevance of the SES framework. We begin by presenting an overview of SES and resilience scholarship, articulating the origins and scope of SES and resilience thinking and identifying the major concepts and most notable authors. Then, we outline a model of change that addresses the two main phases of an adaptive cycle: saturation and reorganization. We use the concepts of rigidity trap and institutional complexity trap to explain the phases of saturation, and draw on the adaptive governance literature to explain the phase of reorganization. In addition, our model of change for SES draws on learning processes as important pathways in reorganization toward adaptation and responsiveness to changing environmental and social conditions. We craft a number of images to capture the various processes discussed and the overall model of change for SES.

*2.1. Social-Ecological Systems and Resilience: Origins and Scope*

Social-Ecological Systems is "*an ecological system intricately linked with and affected by one or more social systems*" [19]. The SES framework further elaborates Ostrom's Institutional Analysis and Development (IAD) and Anderies and colleagues' ones [10,19,20]. It was intended to specify the biophysical aspects of the IAD framework [10]. The IAD framework has been updated through a rearrangement of the list of relevant attributes of governance systems to make it applicable to broader policy settings [9]. Some emerging research aims to combine or integrate the IAD and SES frameworks to help overcome their individual limitations (SES framework as too static and the IAD framework as underspecified) and offer a more powerful framework that allows for deeper multilevel analyses of dynamic changes in social-ecological circumstances by making direct connections between institutional changes and social-ecological outcomes [21].

The SES perspective is closely linked to resilience thinking [22,23]. While the SES framework relates to the structure of a system, resilience focuses on evolution. Social-Ecological System and resilience thinking conceive systems as moving across stability domains according to internal and external disturbances. Key variables of interest are interlinkages among the components of the system and the adaptation of the later according to its resilience. We adopt the social-ecological resilience thinking [24]. Resilience is the magnitude of disturbance that can be absorbed before the system changes its structure by changing variables and processes that control behavior [24]. Social-ecological resilience extends the ecological resilience [25] to the social world and distinguishes from engineering resilience [26]. Engineering resilience considers changes as predictable and focuses on stability near to steady-state equilibrium. The key variable of interest is the recovery time, i.e., the speed of return to the steady-state equilibrium after a disturbance. In that perspective, the system does not change its structure.

The SES framework and resilience thinking together provide a consistent set of concepts for the analysis of co-evolution between humans and nature. Consequently, most researchers use these frameworks as a "*checklist of concepts from which they could qualitatively or quantitatively measure their variables for the sake of case descriptions, case comparisons, or statistical analysis*" ([10], p. 243). To move toward more explanatory power, there is a need for internal theoretical models, especially regarding evolution and change [27,28].

*2.2. An Institutional and Cognitive Model of Change*

2.2.1. Evolution through an Adaptive Cycle

With a better understanding of SES and resilience thinking as a framing, we now propose our model of change, beginning with the adaptive cycle. The global dynamics of SES follows an adaptive cycle that frames the co-evolution between humans and nature [22,23]. The adaptive cycle consists of a destruction–creation process that drives the potential and the connectedness of the SES (Figure 1). SES's potential and connectedness increase within a stability domain until saturation, i.e., occurrence of non-absorbable disturbances. Then, the SES adapts and reorganizes to move toward a new domain of stability. Two phases form the pattern of any adaptive cycle: the saturation and the reorganization phases. The first is a phase of growth until a turning-point where the SES becomes unstable because it is saturated. Mostly, this turning-point corresponds to a regime of over-exploitation without any additional or external stocks of resources. The next phase is marked by a release and a reorganization of the SES toward a new stability domain. After that, a new adaptive cycle starts.

The adaptive cycle heuristics is widely used in SES analysis, but lacks internal explanations of the patterns and triggers of change. To address this, we propose a model of change for SES that offers a theoretical background to the two main phases of an adaptive cycle—saturation and reorganization. We use the concepts of rigidity trap (i.e., a trend toward organizational status-quo) [27,29] and institutional complexity trap (i.e., a trend toward institutional inefficiency) [30,31] to explain the phase of saturation. We draw on the adaptive governance literature to explain the phase of

reorganization, elaborating from four ideal-types of reorganization pathways [32] and the role of learning processes [33–35].

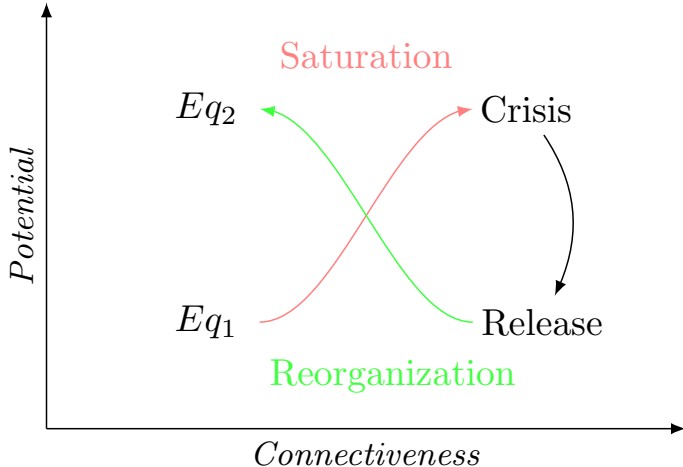

**Figure 1.** Heuristic scheme of an adaptive cycle.

### 2.2.2. Explaining the Saturation of a Social-Ecological System

The rigidity trap reflects the equilibrium where the SES's internal controls lock the system into a pathway that reinforces undesirable outcomes [27,29]. It is a dynamic leading to an organizational status-quo which produce a vicious circle. It is a particular setting where the refinements of the management model and of the institutional design have organized exploitation of the resource characterized by a low natural diversity and a low variation in resource dynamics to maximize the economic use of the resource [23,36]. The governance design appears robust because institutions are highly connected, self-reinforcing, and inflexible, which constrains the innovation ability of the SES [36]. Even if the outcome reveals to be undesirable, the SES is unable to explore any alternative patterns, increasing its vulnerability to disturbance. As a result, the system cannot evolve anymore within the same stability domain and is likely to experience a structural crisis [27,29,36]. For example, a rigidity trap may be a SES organizing an ecologically clean over-exploitation of the resource, which is frequent when public policies do not distinguish the resource from its uses [37,38].

The institutional complexity trap concept puts forward that the extent of the regulatory scope of a given SES has a decreasing marginal impact on its coordination efficiency [30,31]. The governance of any environmental resource is a regime of multiple public policies and property rights specific to different uses of the same shared-resource [38,39]. The regime efficiency (integration) to govern behaviors depends on the scope of regulated uses (extent) and the coherence of the public policies and property rights that structure the regulation.

Together, extent and coherence increase the integration of the regime, and thus the potential efficiency of the governance design. This relation is not linear, the marginal impact of extent on integration is decreasing (Figure 2) [31]. Indeed, the increase of the regime extent impacts negatively on its coherence because of institutional overlaps and frictions. As an illustration, water uses are regulated by policies specific to agriculture, environment, urban planning, and energy, among others; however, these policies could be conflictual or not consistent while overlapping. These overlaps correspond to transversal transaction costs that reduce the efficiency of the coordination and lock the SES in a state of non- feasible additional progress under the same structural conditions. Indeed, the regime is no longer able to coordinate additional trade-offs between different uses and policy objectives [40,41] because transversal transaction costs limit the overall coherence of the regime. The SES becomes vulnerable to disturbances, opening the room for a release and marking the end of the

saturation phase. Over time, expertise in governance stresses the increase of transversal transaction costs by developing siloization and uncoordinated extension of the regime [42].

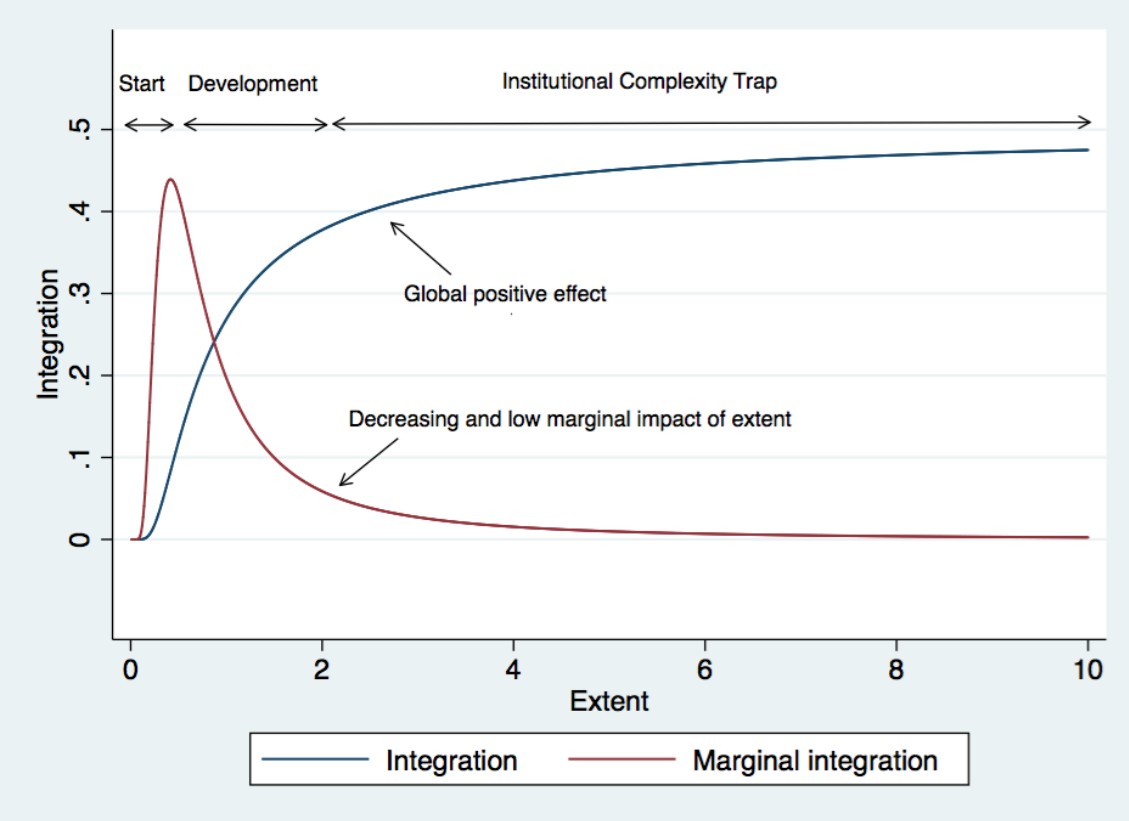

**Figure 2.** The saturation process through an institutional complexity trap [30].

The rigidity trap produces a lack of adaptive capacity and the institutional complexity trap constrains the ability to refine coordination mechanisms. Taken together, these traps drive the SES to a state where it is robust to shock but not necessarily resilient anymore. The SES is vulnerable to disturbances and likely to know a structural crisis, entering into Phase 2 of an adaptive cycle.

### 2.2.3. Explaining the Reorganization of a Social-Ecological System

Phase 2 of the adaptive cycle is the reorganization, including a release and a reorganization process [23]. Social-Ecological Systems reorganize following the patterns of four ideal-types: integrative-synchronous regime, entrepreneurial-synchronous regime, resilient-synchronous regime, transformative-synchronous regime [32]. Each ideal-type has its logic of action and structuring principles depending on the nature of extension (control vs. self-determining) and the goal of the reorganization (poietic vs. dissipative) (Table 1). Extension is the process of producing new regulations, it is "by control" when led by an administrative initiative, e.g., a public policy, and "by self-determination" when actors autonomously create new regulations, e.g., private contracts. These four pathways draw ideal-types, consequently the observed pathways should be hybrids of these.

**Table 1.** Ideal-types of reorganization patterns (adapted from [32]).

|  | **Autopoietic Cohesion** | **Dissipative Cohesion** |
|---|---|---|
| Extension by control | Integrative synchronous regime | Entrepreneurial synchronous regime |
|  | Logic of action: integration | Logic of action: effectuation |
|  | Structuring principle: integrity | Structuring principle: subsidiarity |
| Extension by self-determination | Resilient synchronous regime | Transformative synchronous regime |
|  | Logic of action: resilience | Logic of action: transformation |
|  | Structuring principle: adaptability | Structuring principle: reflexivity |

Integrative-synchronous regimes are problem-solving oriented, and change remains under control. The logic of action is to integrate the SES to reduce the misalignment between the different levels and scales of the SES [43]. Even if this pattern implies changes in the structure of the SES, integrity remains an important goal. Entrepreneurial-synchronous regimes are problem-solving oriented and evolve according to iterative change of components of the SES following a logic of effectuation that seeks to minimize losses [44]. It is an organized form of micro-trial–error learning. Actors impulse reorganization in resilient synchronous regimes by seeking a new balance of the SES functions through reframing problems [45]. The logic of action is resilience, and adaptability of the structuring principle. Transformative synchronous regimes follow the most significant change, regarding structure, by reducing the role of control and reframing problems. Reflexivity is the structuring principle of this change.

In regard of these four ideal-types of reorganization patterns, there are critical questions arising from vulnerability research that speak to the desirability of particular system states with reorganization that address matters of societal costs and benefits around exposures, shocks and stressors and the ability of particular individuals and communities to cope and adapt [17,46]. It emphasizes that designing responses to stress is a trade-off between increasing the resilience of an SES and reducing its vulnerability to shocks [47,48]. Using the resilience perspective, we focus on how SESs could reorganize to increase their adaptability and emphasize the role learning.

In environmental governance, polycentric governance systems, involving multiple overlapping centers of decision-making interacting within an overarching set of rules, are thought to help address the complex interrelationships within our social and environmental systems [49]. Polycentric governance systems exhibit enhanced adaptive capacity [50,51], adapting by changing rules and behavior as they gain experience. Learning can help support a governance system's adaptive capacity [52].

Learning is an important pathway toward adaptation and responsiveness to changing environmental and social conditions. Rules structuring an environmental governance process can enable or constrain the institutional work of learning, and serve as a critical pathway toward institutional change [33–35]. Openness in boundary, scope, information, and choice rules may support learning across a variety of contexts and institutional arrangements, but attention is needed to the different issue contexts and types of learning when thinking about specific design features [33]. The deliberate creation of forums, for example, can bring together decision-makers to share information and learn [53]. However, beyond the structure of the institutional design itself, the personal interrelationships and communication patterns, or social dynamics, can play a key role in promoting and inhibiting learning [54]. Although a deliberate push for polycentric resource management can help encourage rule experimentation, some researchers found that it may not necessarily encourage adaptation and learning [55].

Different types learning can influence reorganization. Serial learning (sequential) differs from parallel learning (simultaneous) and it is noteworthy that the source of learning can be endogenous or exogenous, i.e., other jurisdictions or policy fields [35]. In addition, the concept of triple-loop learning allows identifying the level of learning, and thus the impact of learning on the SES

reorganization [34]. Single-loop learning is incremental; it proceeds in the same pattern of SES dynamics (integrative-synchronous and entrepreneurial-synchronous regimes). Second-loop learning leads to question the cause–effect relationship between components of the SES (resilient-synchronous regime). Triple-loop learning is paradigmatic change and allows for deep transformation and reorganization of the SES toward a new equilibrium (transformative-synchronous regime). The challenge to reorganize an SES is to develop channels for triple-loop learning.

Learning can support innovation in the SES as its implementation reorganizes the SES and proceeds through institutional crafting. Actors use the learning in problem-solving perspective to improve governance and decision-making [56–58]. Learning can result in a change of the structure of the linkages between resource uses and institutions. The network turns from a closed-loop—characterized by rigidity and institutional complexity traps—to more open ones [27,28,33] ("*Characteristics of a rigid system include very few key nodes with a high concentration of influence, and low diversity both in nodes and pathways*" ([27], p. 4)). Consequently, reorganization of an SES necessitates triple-loop learning in association with an opening of the network of uses, resources and institutions.

Figure 3 synthesizes our institutional and cognitive model of change of an SES. The attraction into a rigidity trap or an institutional complexity trap saturates SES over the years. The SES becomes vulnerable to external disturbances and is likely to experience a crisis. The crisis opens a phase of release that ends-up with the reorganization of the SES. The reorganization consists of embedded pathways of change (multiple spatial and time scales) and is driven by individual learning, innovation, and diffusion. These three drivers allow for a paradigmatic shift of the SES structure, opening the network of actors. It defines a new equilibrium with a more balanced relationship between ecological and social systems, and with more flexibility and adaptability.

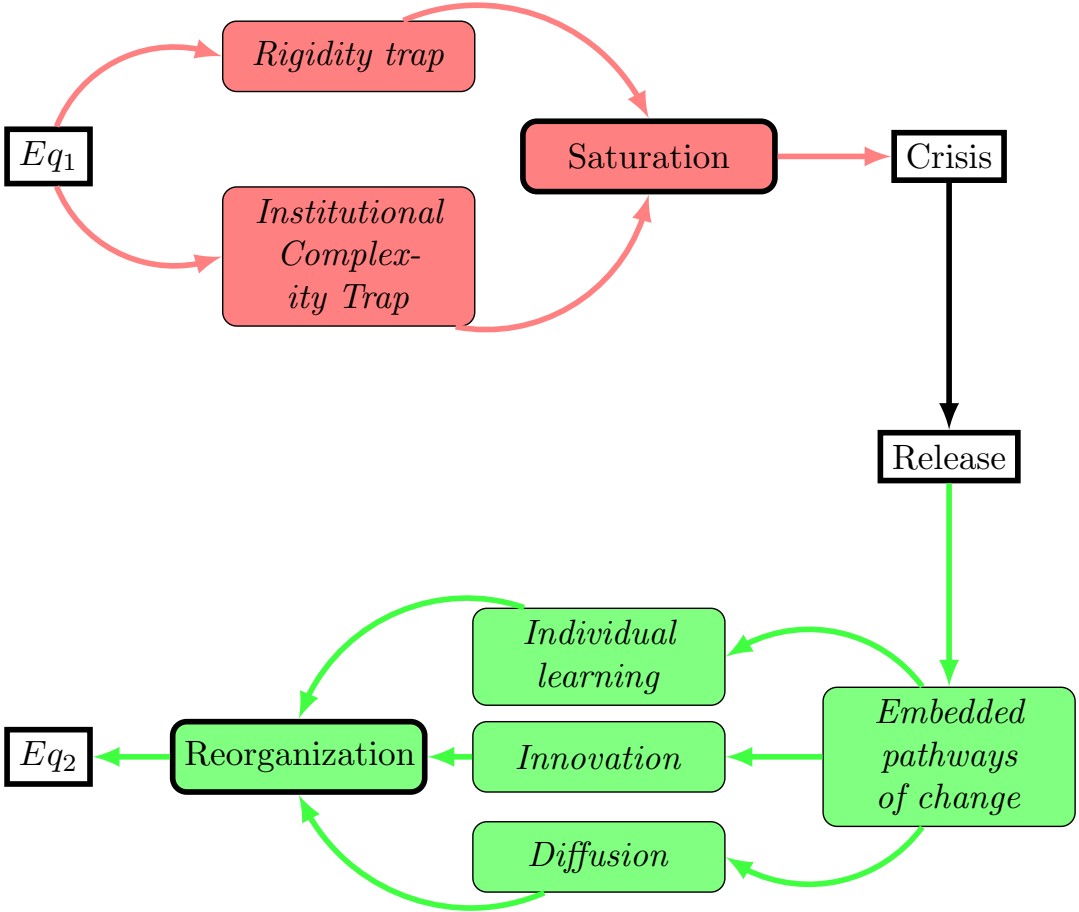

**Figure 3.** An institutional and cognitive model of change for SES.

## 3. SES Evolution and Measurement: Insights from the Case of Water Security

In Section 2, we propose a model to understand the dynamics of SES, which is crucial in a context of global transformations. In this section, we elucidate key methodological limitations that prevent reliable tests of the institutional and cognitive model of change of SES and knowledge accumulation. We put forward recommendations to tackle these methodological issues. We use the case of water security as an illustration of SES measurement. Water security makes for a good illustration given its high-priority in the political agenda, its fundamental social-ecological essence, and its multi-level and multi-uses perspective.

*3.1. Limitations to General Knowledge about Water Security: Measurement Heterogeneity and Time-Limited Analyses*

Over the past decade, water security has emerged as a new currency of the international water community [59]. It can broadly be seen as an extension of sustainable development thinking to water resources with the focus on the quantity and quality of water supply for societal and ecological needs [3]. There is an extensive and growing body of scholarship examining the adoption of and application of water security in a wide array of journals and from a diverse set of authors across diverse geographic regions and scales around the world [60,61]. In bridging both the environmental and social spheres, water security can be seen as a tool for the adaptive management of an institutional water regime and contributes to improved governance [2]. However, knowledge accumulation is limited in that area of research.

We argue that there are three primary factors that limit knowledge accumulation around water security and, thereby, hinder its ability to serve as a tool for adaptive management of SESs: large diversity of methodology, case specificity, and cross-sectional studies. The two first limitations prevent comparability and thus the external validity of findings. The former is a matter of internal validity: dynamic mechanisms are not properly observed or tested, which prevents inferential and causal claims.

The first limitation that prevents knowledge accumulation around water security is heterogeneity of measurement of water security. There has been an explosion of assessments of water security. The analysis of the 124 empirically-based studies recorded in Scopus between 2010 and 2015 highlights that indicators to operationalize water security vary considerably around the world [60]. The authors report the use of 15 sub-indicators, and that the majority of the studies use only two of them: sufficient quantity (91% of the studies) and acceptable quality (51% of the studies). This large methodological diversity is also characterized by a diversity of conceptual frameworks, which serves to limit comparison. Indeed, they are generally characterized by a high subjectivity which can reduce the relevance of their normative use [2]. For instance, water security scores of Vietnam, Thailand, the Philippines, and Kyrgyzstan are contradictory in the literature, being good or weak depending on the study [2]. This heterogeneity is consequential considering the current evidence-based policy-making paradigm. What does "evidence-based" mean regarding such an extensive range of appraisal? In addition, it actively prevents the growth of a common scientific knowledge because opposing measures of common observation units are shared with the same label—without mentioning this diversity.

The second limitation is case specificity in terms of how we study and understand water security. There is a growing body of place-based research that investigates how water security is adopted, engaged, and implemented in various environmental and social contexts around the world. Twenty-one percent of peer-reviewed studies published between 2010 and 2015 dealing with water security are place-based cases [60]. The persistent diversity in perspectives and applications of water security suggests that scholars adapt the concept to the contexts of the cases they are studying. In the same sample of studies, 60% of the studies using a definition of water security offer a new definition, and thus are case-based [60]. While this can suggest the need for a more community-context view to understand and realize water security, it might also suggest a limited generalizability of water security and hinder our knowledge of water security.

Finally, the third limitation of knowledge accumulation around water security focuses are static, or measuring cross-sectional, meaning that samples cover only one period of time (mostly a year). It prevents the study of water security evolution, and of SESs more generally. As an illustration, a significant strength of water security in comparison to Integrated Water Resource Management (IWRM) and nexus—competing concepts—is that water security is thought as a lever of development and transitions [2,4]. Integrated Water Resource Management and nexus consist mostly of refining an existing situation without any systemic change in most cases [62,63]. Integrated Water Resource Management and nexus-based governance concepts are prone to institutionalizing the saturation of the SES. Regarding our institutional and cognitive model of change, we could assume that they frame integrative and entrepreneurial synchronous regimes mainly. Existing datasets prevent testing this core assumption, which impedes analyzing the impact of social reforms on the ecological outcome of an SES.

*3.2. Dealing with Measurement Disparity and Evolution: Recommendations to Reinforce External and Internal Validity*

To overcome methodological heterogeneity and case-specificity that constrain external validity of studies, we recommend: (i) enlarging the number of observations; (ii) carrying out a replicable empirical strategy; and (iii) including a time dimension. First, increasing the number of observations and cases automatically increase the external validity as it reduces the case-dependency of results. In addition, it makes it easier to identify the strategy and delineate impacts of contextual variables

(climate, economic activity, etc.) from effects of explanatory variables (learning processes, institutional setting, resource uses, etc.). Seven prominent databases that contain SES or commons related datasets exist. Recommendations have been formulated to favor their use in a replicable and consistent manner [11]. These datasets cover about 2000 cases, and coding strategies are all based on the SES or IAD framework, which limits inconsistency among studies. (Two important meta-database initiatives include: (1) SESMAD, 2014. Social-Ecological Systems Meta-Analysis Database: Background and Research Methods. Available from: http://sesmad.dartmouth.edu/; and (2) Resilience Alliance and Santa Fe Institute. 2004. Thresholds and alternate states in ecological and social-ecological systems. Resilience Alliance. (Online.) http://www.resalliance.org/index.php/thresholds_database). The literature on water security should gain in developing this kind of common research practice and data sharing.

Second, replicable research design is a necessary condition for expending datasets over time, by adding new observations or variables and by facilitating merging dataset. The choice of raw data and coding strategy are critical. We recommend using data that are reliable, free and "protected" to ensure their accessibility over space and time. Coding strategy contributes to reach this goal. It should be clear, not case-dependent and easy to replicate over many observations. In this regard, automated-coding looks necessary. It is widely used to analyze the resource and its characteristics but remains rare and difficult in the study of institution and actor relationships. Recent methodological progress in social sciences enables (semi) automated-coding of institution–actor attributes. For instance, a dataset links actor, rule, and the deontic categories of Ostrom's institutional grammar [64] to study polycentricity across 11 Colorado oil and gas regulations [65]. In that way, they produce a convincing measure of polycentricity and other network attributes that could be transposed to the analysis of the rigidity trap. Discourse network analysis [66] techniques could be used to appraise the degree of institutional innovation and its diffusion to depict the logic of actions SES pathways of change [32] and reorganization of SESs. Finally, inferential network analysis [67] methodologies permit tests of causality between these institutional characteristics and actor attributes with environmental outcomes in a pervasive manner. Geographical factors are of primary importance in an SES [28]; similarly, space and scale play a critical role in water security [60]. Consequently, SESs studies must have a spatial dimension and ideally include GIS information into databases. It is decisive to identify central actors, rules, context path-dependencies and scale-related non-linearity.

Finally, analysis of change necessitates including time-related variables such as date, time lag, and event history. While it seems to be commonplace, it is not a trivial challenge. Methodologies to study ecological patterns in SES exist [68–71] but measuring and observing social components over the years remains difficult. Consequently, very few studies manage change over time. The first attempt to model feedback between water security, risk and economic growth is very recent [4] and researchers started to draw and share lessons from their experience in modeling of coupled human–nature systems [12]. ("The following eight lessons were identified that if taken into account by future coupled human–natural-systems model developments may increase their success: (1) leverage the power of sensitivity analysis with models, (2) remember modelling is an iterative process, (3) create a common language, (4) make code open-access, (5) ensure consistency, (6) reconcile spatio-temporal mismatch, (7) construct homogeneous units, and (8) incorporating feedback increases non-linearity and variability" ([12], p. 896).) It is consistent with our previous recommendations: common language, open-access code, considering spatiotemporal mismatch, and homogeneous units are indispensable. Transposing qualitative materials into quantitative materials to study water resource regimes extent, coherence and integration over centuries seems to be a fruitful avenue [30]. Coupling these types of analyses with environmental outcomes seems promising to the analyses of SESs evolution and the identification of global changes as exogenous shocks. Consequently, we recommend including a time-variable in each study even if the research is time invariant. This would facilitate future aggregations of existing studies and enable dynamic analysis of SES evolution.

## 4. Earth Observations for Water Security: The Data Cube

We identify three primary methodological limitations to testing our institutional and cognitive model of change of SES. The case of water security studies highlights significant issues related to measurement heterogeneity, case specificity and static perspective. Data cube technology could help overcome current limitations and offer reliable avenues to test hypothesis about the dynamics of social-ecological systems and water security by combining spatial and time data with no major technical requirements for users.

### 4.1. Earth Observation Data to Consider the Spatial Dimension of Social-Ecological Systems and Water Security

Remotely-sensed Earth Observations (EO) data appear to be necessary materials to take into account the spatial dimension of water security, and SES in general. Acquired by satellites together with in-situ measurements, they are a source for effective, quantitative and integrated capabilities (e.g., social and ecological) to monitor trends and variability for social-ecological systems. Regarding water security, authors already scrutinize water quantity and quality [6,72]. With EO data, one can map and monitor water bodies. Optical and radar sensors help identifying changes in area and water quality information can be obtained by applying algorithms to retrieve information (e.g., total suspended matter and chlorophyll content) from water color [73]. Usually, spatial resolution is about 10–30 m and temporal resolution is about 5–15 days.

One of the main advantages of using EO data is that they provide information at the pixel level and therefore can be aggregated or disaggregated according to ecological or social variables to ensure the spatial fit of the components of an SES. For example, a network of actors can be represented following its spatial perimeter and then can serve as the basic statistical unit from which pixel values can be aggregated and various statistical information can be derived using zonal statistics. Such an approach suits the analysis of spatial mismatch between actors, institutions and resource [74], but with large N dataset which allows inferential analysis. Consequently, the analysis is not framed by administrative boundaries but by the footprint of the variables of interest. In addition, this strategy of data collection could be used for treatment effect research design by allowing to test the treatment in space or time. As an illustration, the impact of the institutional complexity trap on resource use could be tested robustly.

Two principal missions allow retrieving EO data related to water security. Firstly, the Gravity Recovery and Climate Experiment (GRACE) mission allows tracking anomalies of the Earth's gravity field and, consequently, extract information on groundwater resources. Secondly, the Soil Moisture and Ocean Salinity (SMOS) mission enables mapping soil moisture and can provide useful information for monitoring droughts and extreme events. Other information can be generated from EO data such as precipitation, evapotranspiration, land cover, snow, and temperature, which are all important variables to monitor the water cycle [72]. Usually, EO data are processed to generate information products that describe real-world variables, and these products are in turn converted into evidence-based information for decision support [75].

### 4.2. The Data Cube to Make Earth-Observation-Based Research Replicable over Time and Space

Measurement heterogeneity and non-replicability are significant limitations to the study of SES dynamics. EO data cubes overcome these limitations. EO data can provide an efficient and effective mechanism to support water security monitoring. However, the full potential information of EO data is difficult to realize mainly because of their complexity, increasing volume, and the lack of efficient processing capabilities [76,77]. Consequently, new approaches are required to rapidly analyze data in a transparent and repeatable manner and to facilitate the transformation of many EO data into actionable information and decision-ready products [75]. Data cubes are part of these new approaches.

To tackle these issues and bridge the gap between user expectations and current big cata analytical capabilities, EO Data Cubes (EODC) are a promising solution to store, organize, manage and analyze EO data. The main objective of EODC is to facilitate EO data usage by addressing volume, velocity,

and variety challenges, and providing access to many spatiotemporal data, for a given geographic area over a specified time period, in an analysis ready format [13]. EODC reveals to be a promising tool for analyzing SES dynamics and testing the institutional and cognitive model of change.

With EODCs, researchers could address the current methodological limitations enabling reliable and causal analysis of SES dynamics. It constitutes a data infrastructure offering: open access materials, systematic measurements and sample combination in a versatile format. Consequently, it facilitates replicable empirical strategies, common measurements and enlarging sample size by including new geographic areas and new periods. Because of the versatility of EODC, EO data can be merged with social data, building an integrative dataset of SES. For instance, with EODC, a discourse network analysis dedicated to grasp degrees of institutional innovation in learning process could be combined with water flows and water quality information over years.

There are four currently operational technologies, namely the Open Data Cube supported by Digital Earth Australia [14], the EarthServer [78], the Google Earth Engine [79], and E-sensing [80]. These initiatives are paving the way to broaden the use of EO data by larger communities of users; they represent about 50 different implementations worldwide supporting decision-makers with timely and actionable information converted in meaningful geophysical variables; and ultimately are unlocking the information power of EO data. For example, the Swiss Data Cube holds currently 34 years of Landsat (1984–2018) and three years of Sentinel-2 (2015–2018) Analysis Ready Data (ARD) over Switzerland [13]. (An essential pre-condition to support user applications and generating usable information products is to facilitate data access, preparation, and analyses. The systematic and regular provision of Analysis Ready Data (ARD) can significantly reduce the burden of EO data usage. To be considered as ARD, data should be processed to a minimum set of requirements (e.g., radiometric and geometric calibration, atmospheric correction, and metadata description) and organized in a way that allows immediate analysis without additional effort. ARD correspond in optical imagery to surface reflectance products). This corresponds approximately to 7000 scenes from 1984 to 2018 for a total volume of 4 TB of ARD and more than 110 billion observations that can be converted in different environmental variables.

Most of these EODC implementations are using Landsat data, partly because it is the oldest EO program, providing continuous observations for more than 45 years. In addition, since 2008, the entire data archive is freely and openly accessible [81]. This creates an unprecedented opportunities to use Landsat data in different contexts such as land cover changes, ecosystem mapping or water monitoring [82,83]. Besides Landsat, the Sentinels program, operated by the European Space Agency (ESA), forms another valuable EO data source. Sentinels are part of a satellite constellation for the operational need of the European EO program called Copernicus [84,85]. These two exhaustive sources favor establishing common, reliable and precise measurements that minimize the risk of missing data over time and space.

Among the major benefits of EODC is that it makes data analysis easier, facilitating data access and distribution and lowering the technical barriers for managing and processing many EO data (Figure 4). Traditionally, satellite imagery is downloaded, processed and provided to users on a custom basis. This process serves a single purpose each time analyzing data locally (e.g., desktop computer) and downloading data on scene-based file. With the paradigm shift advanced by EODC information becomes available more rapidly, the burden of data preparation and usage is drastically reduced [14,86]. Moreover, the data quality is enhanced as ARD are homogenous. They are combined in a consistent workflow offering pre-processed data with the same procedure along the entire time-series. Similarly, it makes easier the cross-checking with other databases because data can be accessed through interoperable Application Programming Interfaces (APIs). This allows seamlessly integrating data coming from various sources into a single analysis in a very efficient and effective manner. Consequently, the widespread availability of EO data, the advent of ARD, and the large processing capabilities provided by high performance computing (e.g., clouds, clusters) allow envisioning supporting policy frameworks such as the SDGs [87]. EODC can provide the long baseline required to

determine trends, define present, and inform future, and thus provide a good fit between the analytical scope of SES evolution and empirical tests.

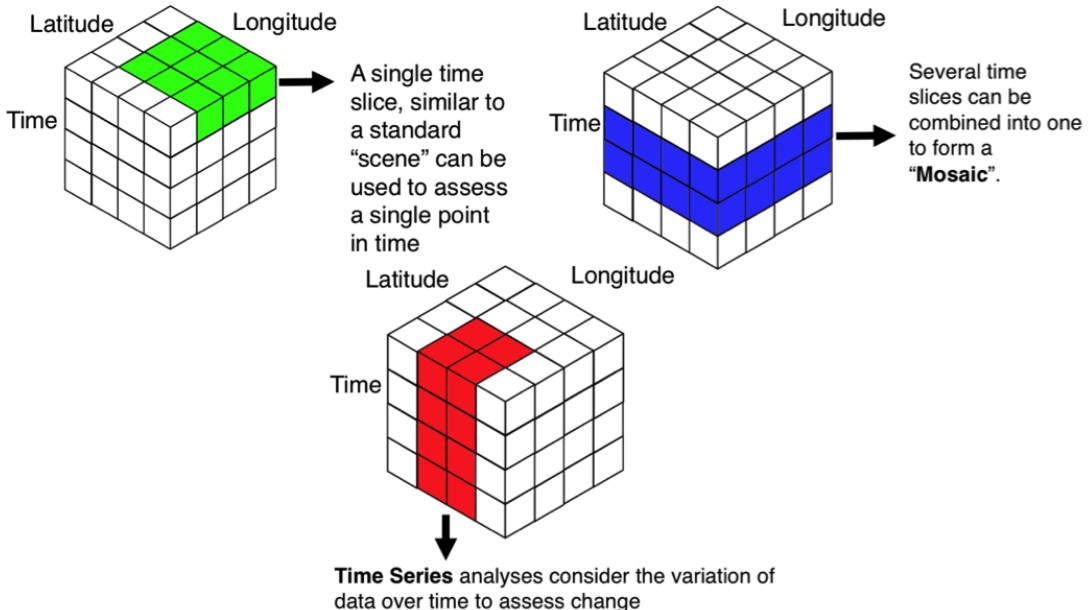

**Figure 4.** Sampling in a data cube (adapted from [88]).

*4.3. Avenues for Implementing an Analysis of Water Security Dynamics through Data Cube*

Currently, the most advanced EODC implementation is represented by Digital Earth Australia (DEA) that already developed some interesting information products related to water management. In particular, using the Water Observations from Space (WOfS), they map the presence and extent of surface water over the entire country using 27 years of Landsat observations [89]. It allows for flood risk assessment, agricultural water tracking, and mangrove monitoring. Similarly, Australia has developed an algal bloom early warning alert system informing SDG 6 (clean water and sanitation). This initiative contributes to build common and replicable measurements of the water security ecological system.

We argue combinations of these measurements of the ecological system with variables about the social system related to water security, namely institutions and actors, is promising. Moreover, there is a momentum for these combinations as social sciences recently developed new methods for collecting information and coding variables about institutions and actors configurations [31,65,66]. For instance, these new methods allow for systematic measurement of polycentricity and policy content with many observations over time and space. It could advantageously be merged with ecological dataset. Then, because the dataset has a time dimension, it becomes possible to analyze patterns of change. For instance, we could appraise the impact of socio-institutional changes in oil and gas regulation [65] on water quality. Regarding the on-going debates about fracking, it would result in substantial added-values for both science and policy-making. More generally, it enables to link fine-grained measurements of polycentricity with water resources and thus to provide new insights into influential water governance paradigms such as the Integrated Water Resource Management.

Regarding water security, EODC paves promising avenues in the use of the Water Security Index (WSI) [5]. The WSI results from a quantitative and integrated (e.g., physical and socio-economic) approach to develop a spatial multi-criteria analysis framework. The index combines four criteria, "availability", "availability services", "safety and quality" and "management", based on the measurement of ten indicators (Table 2). Indicators about water availability are at the 0.5° spatial resolution while the other indicators are at the country or river scales, and the time coverage excludes dynamic analysis.

**Table 2.** Composition of the water security index (adapted from [5])

| Security Criteria (Weights) | Indicators (Weights) | Spatial and Temporal Scales |
|---|---|---|
| Availability (45%) | Water scarcity index (70%) | 0.5° spatial resolution; |
| | | Monthly mean value of 2010 |
| | Drought index (15%) | 0.5° spatial resolution; |
| | | Monthly mean value of 2012 |
| | Groundwater depletion (15%) | 0.5° spatial resolution; |
| | | Monthly mean value of 2010 |
| Accessibility to water services (20%) | Access to sanitation (40%) | Country scale; 2014 |
| | Access to drinking water (60%) | Country scale; 2014 |
| Safety and quality (20%) | Water quality index (50%) | Country scale; 2012 |
| | Global flood frequency (50%) | Country scale; 1985–2003 |
| Management (15%) | World governance index (70%) | Country scale; 2010 |
| | Transboundary legal framework (15%) | River Basin scale; 2015 |
| | Transboundary political tension (15%) | River Basin scale; 2015 |

A first avenue is to take advantage of EODC versatility to replicate the measure of the WSI and expand its time coverage to several decades. It would allow observing changes in water security situations and in relations between the criteria of WSI (e.g., accessibility to water services with safety and quality). Then, the longitudinal measure of the WSI could be combined with accurate measures about actors and institutions, i.e., under the country scale. Because EODC facilitates spatial matching thanks to disaggregating–reaggregating pixels procedures, one could test our institutional and cognitive model of change for SES (Figure 3). For instance, it allows estimating the association of new regulations with WSI changes to grasp the institutional complexity trap dynamic. This would allow identifying the level of saturation of SESs (see Figure 1).

The potential contribution of EODC for water security is wider than expanding the use of the WSI. EODC allows measuring an index that corresponds more with the water systems and current use of the resource. For instance, ecological structure is a critical component of any SES. With EODC, it is possible to characterize the river degree of fragmentation and the water extent. EODC allows observing total suspended matter and chlorophyll content in water bodies which will enable for appraising water quality in the past (where no sampling have been made) and in the future reliably.

EODC offers accurate measures of water-related hazards, for example, to observe floods and droughts events and to determine human and economic exposure by crossing the spatial extension of events with population and economic datasets. A concrete illustration is the mapping of water extent dynamics across the entire Australian continent [89]. Using EODC, the mapping combines observations of permanent and temporary water bodies at 30 m resolution, with a temporal resolution of 15 days during 27 years. Such information gave insights on presence and frequency of occurrence of surface water across various spatial scales ranging from a few tens of meters about to the entire continent. It gave key information about spatial and temporal extents of significant stream flows and floods. Another example is the PreView Global Risk Data Platform using more than 30 years of EO data to map floods and droughts at the global scale [90]. It allows determining human and economic exposure as well as associated risks [91]. This can give essential information on socio-economic aspects related to natural hazards.

These EODC tools could be used to set up experimental research design, and use causal modeling to test our model of change for SES. For instance, the institutional complexity trap has been studied in the case of flood policy in Switzerland by showing an unexpected increase in institutional overlaps and conflicts that should limit regulation efficiency [31]. With the PreView Global Risk Data Platform, we could link this increasing complexity of the Swiss flood regulation with exposure to floods. Learning

could be tested as well. Since the 1960s, changes in experts influence significantly impacted Dutch and American flood governance in order to limit exposure to floods during [92]. Linking these changes with accurate measures about floods, through EODC, allows for a systematic analysis of the reorganization of SESs.

These avenues highlight that the versatility of the EODC enables research designs with treatment effect, and therefore causal inference. Indeed, natural experiments are rare in the social world and ecological information are not reliable or accessible worldwide and for every time. Consequently, elaborating robust experimental designs for SES analysis is extremely difficult. With EODC, if we observe a change in relevant social variables (e.g., new policies, new information initiating or diffusing learning), we could design robust causal tests that would advance empirical and theoretical knowledge about SES dynamic.

## 5. Conclusions and Next Steps

In this paper, we propose a model of change for SES that explains how institutions and natural resources systems co-evolve. In doing so, we offer a more dynamic understanding of SES that stands on three causal mechanisms: institutional complexity trap, rigidity trap, and learning processes. We rely on the adaptive cycle heuristic and focus on the saturation and the reorganization phases of a SES. We discuss how rigidity traps and institutional complexity traps drive the saturation of an SES until a crisis. Rigidity traps reduce adaptive capacity (i.e., the ability to explore alternative patterns) while institutional complexity traps reduce the ability to refine coordination mechanisms (i.e., the marginal effectiveness of new institutions). After a crisis, four ideal-types grasp the SES's diversity of reorganization patterns. We discuss how transformative and resilient synchronous regimes are more likely to frame a rebound. We argue that learning processes are pivotal in defining the ideal-type of reorganization pattern that the SES is likely to follow. Learning mechanisms and capacities contribute to determining the level of innovation and its scope of diffusion in a reorganizing SES.

This model of change offers many testable propositions, however there are three main methodological limitations that prevent hypothesis testing and results-comparison. We highlighted these limitations using water security as an illustrative case. These limitations include: heterogeneity of measurement, case specificity and static perspective. These methodological limitations bound the external validity and reproducibility of research designs that address SES dynamics. We suggest three primary recommendations to overcome these limitations: (1) increase number of observations and the geographic scope; (2) provide replicable measurements regarding both data availability and method transparency; and (3) include time variables in the analysis—even in the case of panel dataset.

Finally, we put forward Earth Observation Data Cubes (EODC) as a promising avenue to address these methodological limitations and recommendations, and thus to contribute to a better understanding of SES dynamics. Earth observations (EO) enable focusing on the spatial dimension of SESs and water security, but they are difficult to perform. Data cubes provide EO through a handy interface, exclude complex technical knowledge from the users, and combine data over the years. They make easier the combination of social and ecological data over many observations and facilitate broadening the space and time coverage, which makes it a unique tool for testing generalizable propositions of SES change. Moreover, data cubes enable the data structure fits with the theoretical unit of analysis. Indeed, the basic unit of EODC is the pixel, but then pixels could be aggregated according to the analytical needs. It means the dataset is less dependent of administrative boundaries but could stick with the resource footprint for instance.

Overall, our contribution to the scholarship and research on SES is threefold. First, we provide a model of change to enable causal hypothesis on the evolution of SES. Second, we identify three methodological limitations preventing reliable and generalizable results on the evolution of SES through an illustration of water security. Finally, we propose new tools through the EODCs to overcome these limitations and enable an easy combination of social and ecological data over space and time.

As next steps, we foresee two primary ways to combine social and ecological datasets. The first way consists of building a georeferencing social network and linking this with ecological system [28]. The second way is to give a spatial perimeter to institutions and to link this with the ecological system. This has recently been done to highlight spatial misfits [74], but it could be used to analyze institutional complementarities, in relation with the institutional complexity trap mechanism, or mechanism of learning and diffusion to studies reorganization patterns in a dynamic way. However, with only four operational data cubes in Australia, Colombia, Switzerland and Taiwan (but with many more appearing (recently, the African regional data cube has been launched covering Kenya, Senegal, Sierra Leone, Ghana, and Tanzania, while other EODCs are under development in Vietnam, Uganda, United Kingdom, Georgia and Moldova and more that 30 other countries have expressed interest, with the long-term objective of having continental or even global coverage)), there remain challenges to data cubes' extensive use. Further, harnessing the full potential of EO data for supporting informed decision-making on water security requires developing new algorithms (e.g., water related indices and aggregations methods) and tailored applications specific to the water security domain. Finally, we invite researchers to share their data in a common and free access repository to accelerate knowledge accumulation by favoring comparison, cross-analysis, and collaboration.

**Author Contributions:** Conceptualization, T.B.; Methodology, T.B., A.K.G.; Writing—Original Draft Preparation, T.B., A.K.G. and G.G.; Writing—Review and Editing, T.B., A.K.G. and G.G.; and Visualization, T.B. and G.G.

**Funding:** This research received no external funding.

**Conflicts of Interest:** The authors declare no conflict of interest.

## Abbreviations

The following abbreviations are used in this manuscript:

| | |
|---|---|
| ARD | Analysis Ready Data |
| EO | Earth Observation |
| EODC | Earth Observation Data Cube |
| IAD | Institutional And Development framework |
| IWRM | Integrated Water Resource Management |
| SES | Social-Ecological System |

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
