# Peer review of "Explaining and Measuring Social-Ecological Pathways: The Case of Global Changes and Water Security"

_sustainability, doi:10.3390/su10124378_

Round 1

Reviewer 1 Report

General comments

The article titled “Explaining and Measuring Social-Ecological Pathways: The Case of Global Changes and Water Security”, can become suitable for publishing in the “Sustainability Journal” only if major changes are made. The development and the application concern a methodology on social-ecological systems and water security, which combines spatial and temporal data and can help policy-makers / water experts to improve their decisions regarding water resources. Overall, the manuscript is sufficiently readable. Concerning the abstract, it is quite informative and reflects the body of the manuscript, however, the Introduction and SES and Evolution: Theoretical Framework needs improvement from insufficient background information. I suggest the authors include the term of vulnerability and how it is connected to resilience. Additional, an important missing point of the manuscript is how the various uncertainties (models, data quality, indicators etc.) are determined objectively and how the best set of tools is evaluated and documented. It is not also very clear the criteria for the selection of the method and how the comparison of the outputs is performed. The section with the proposed methodology must also be improved. Furthermore, I would like to see an extra session 4.4 that will describe a hypothetical Central Earth Observation Data Cubes with the four operational Data Cubes and that can improve or integrate new or existing databases with various available data (social, environmental, political etc.) and how will connect new databases in this integrated data cube system. After this analysis, the results will be more stable especially in the lines 488 – 498. Last but not least, I would like to see an analysis for strategies of Ecosystem Stewardship and how incorporates with the overlapping approaches to sustainable development and the role of vulnerability in these systems. Finally, I suggest describing specific indicators for water security and general for water resources management (drought, drought vulnerability, floods, flood risk assessments etc.) and how these results can help the policy-makers.

Specific comments

Line 9: Please consider the “hypothesis testing” instead of "test hypothesis". Also, please check the same term in the manuscript.

Line 26: Please change the “are” with “is”.

Lines 56 -57: Please add reference/s in this sentence.

Line 60: Please consider “of” instead of "about".

Line 86: Please delete the (p. 33) there is the reference and it is harmonized with the journal’s standards. The same issue with the line 93, (p. 243).

Figure 1: I suggest the authors adapt the current graph in their methodology, thus, the potential reader will understand better the theoretical approach including the two main phases of adaptation.

Line 164: Please change the "seek" with “seeks”.

Line 180: Please check the reference, also line 313

Figure 3: I think it is essential for transformation to be integrated in this flowchart.

Line 240: There is a missing preposition after the word "seen a tool" I suggest to change it "seen as a tool" maybe?

Session 4.2: I propose the authors include the quality of available data and the possibility of cross-checking different databases.

Line 406: Please change the word "currenlty" to “currently”.

Lines 413 - 417: Please use reference/s

Line 443: Please consider the word "merge" I propose to change the phrase be merged or be merging.

Lines 444 - 451: The authors need to be more specific with these systems. For example, they called the WSI but there isn't the path of these calculations. Additionally, the authors wrote the following "They measure water security as a function of ‘availability’, ‘accessibility to services’, ‘safety and quality’, and ‘management’. " My question is that how they measure this concept?

Author Response

Response to Referee Number 1

We are extremely grateful for your positive view of our paper and, more importantly, to the many constructive criticisms you made. We have found your comments appropriate and helpful and believe that they have allowed us to improve the content and exposition of our argument substantially. Please find below our detailed responses to your specific concerns (your original comments are in italics).

General comments

The article titled “Explaining and Measuring Social-Ecological Pathways: The Case of Global Changes and Water Security”, can become suitable for publishing in the “Sustainability Journal” only if major changes are made. The development and the application concern a methodology on social-ecological systems and water security, which combines spatial and temporal data and can help policy-makers / water experts to improve their decisions regarding water resources. Overall, the manuscript is sufficiently readable.

Concerning the abstract, it is quite informative and reflects the body of the manuscript, however, the Introduction and SES and Evolution: Theoretical Framework needs improvement from insufficient background information.

Thank you for these remarks, We have revised the manuscript extensively and accordingly by adding more background information within all along these sections. For instance, in this version concept such as extension is clearly defined (see lines 174-177). We have also elaborated further on central theoretical propositions to make them clearer, like for our contribution to the adaptive cycle (lines 122-128).

I suggest the authors include the term of vulnerability and how it is connected to resilience.

Thank you, we have taken fully into account this suggestion and have used that to clarify our approach (lines 191-197). Indeed, vulnerability and resilience could be considered as the sides of the same coin. This suggested link to vulnerability allows reminding the role of risk and disturbance exposure in SES.

Additional, an important missing point of the manuscript is how the various uncertainties (models, data quality, indicators etc.) are determined objectively and how the best set of tools is evaluated and documented. It is not also very clear the criteria for the selection of the method and how the comparison of the outputs is performed. The section with the proposed methodology must also be improved. Furthermore, I would like to see an extra session 4.4 that will describe a hypothetical Central Earth Observation Data Cubes with the four operational Data Cubes and that can improve or integrate new or existing databases with various available data (social, environmental, political etc.) and how will connect new databases in this integrated data cube system. After this analysis, the results will be more stable especially in the lines 488 – 498.

Thank you very much for this extremely useful comment, we have clarified these methodological concerns. The EODC is a tool that facilitates data collection in an automated and replicable manner, based on the use of freely accessible earth observation data. The output of EODC is Analysis Ready Data which could be the input of any numerical or statistical modeling. We have clarified this chain and emphasized that it is a source of reliability as code are available, and interface exists to create interoperability (lines 452-457, and footnote 5). Besides, we mentioned two important meta-database initiative and shared good practices in modeling and using these datasets with a large number of observations (lines 320-322, lines 349-352, footnotes 3 and 4).

In the previous manuscript, we did not mention the interfaces that allow interoperability. These interfaces make a combination of Data Cube more efficient than a central one: it is more versatile, it limits the risk of significant decrease of research capacity because costs are shared, and it favors multiple initiative and fast development as each Data Cube could be autonomous and update the others. Because of these reasons, we did not elaborate on a central Data Cube.

Last but not least, I would like to see an analysis for strategies of Ecosystem Stewardship and how incorporates with the overlapping approaches to sustainable development and the role of vulnerability in these systems.

We have considered carefully this suggestion of yours and we found it truly useful. While in this paper we could not directly elaborate its implications, and we could not explore more in-depth the overlaps between existing approaches to sustainable development, we have highlighted the novelty of the use of Data Cubes and specified its contribution to water security.

 Finally, I suggest describing specific indicators for water security and general for water resources management (drought, drought vulnerability, floods, flood risk assessments etc.) and how these results can help the policy-makers.

This comment of yours proved to be extremely useful as it helped us to better formulate our methodological contribution. We have described extensively specific indicators for water security, the Water Security Index (Gain et al., 2016), and showed how EODC could expand it (lines 485-501, table 2). We have generalized this to water resource management and implications for policy-makers (lines 479-483 and lines 503-521).

Specific comments

Thank you very much for these most useful specific comments. We have revised the paper accordingly.

Line 9:   Please consider the “hypothesis testing” instead of "test hypothesis".   Also, please check the same term in the manuscript.

(Line 6) Changes has been made accordingly

Line 26: Please change the “are” with “is”.

(Line 26) Thank you very much, this mistake   has now been correct

Lines 56 -57: Please add reference/s in this sentence.

(Line 62) References have now been added

Line 60: Please consider “of” instead of "about".

(Line 65) This mistake has now been correct

Line 86: Please delete the (p. 33) there is the reference and it is harmonized   with the journal’s standards. The same issue with the line 93, (p. 243).

References   are now correct

Figure 1: I suggest the authors adapt the current graph in their methodology,   thus, the potential reader will understand better the theoretical approach   including the two main phases of adaptation.

Figure 1 Thank you very   much for these most useful comments. We have revised the figure extensively   and accordingly.

Line 164: Please change the "seek" with “seeks”.

(Line 184) This mistake has now been correct

Line 180: Please check the reference, also line 313

Thank you, we have checked these references

Figure 3: I think it is essential for transformation to be integrated in this   flowchart.

Figure 3 Thank   you we have changed extensively the figure 3

Line 240: There is a missing preposition after the word "seen a tool"   I suggest to change it "seen as a tool" maybe?

(Line 264) Thank you, this mistake has now been correct

Session 4.2: I propose the authors include the quality of available data and the   possibility of cross-checking different databases.

(Line   452-457) Thank you, we have clarified this

Line 406: Please change the word "currenlty" to “currently”.

(Line 406) This mistake has now been correct

Lines 413 - 417: Please use reference/s

(Line 436-444) Thank you we added references to this passage.

Line 443: Please consider the word "merge" I propose to change the   phrase be merged or be merging.

We have extensively modified all the section   4.3 and avoided this mistake.

Lines 444 - 451: The authors need to be more specific with these systems. For example,   they called the WSI but there isn't the path of these calculations.   Additionally, the authors wrote the following "They measure water   security as a function of ‘availability’, ‘accessibility to services’,   ‘safety and quality’, and ‘management’. " My question is that how   they measure this concept?

Thank you very much for these most useful comments. We have revised the   paper extensively and accordingly.

(Lines 485-501 and table 2) This version includes a detailed explanation of   the Water Security Index and how data cube could contribute to a better   measurement.

Reviewer 2 Report

This paper, entitled “Explaining and Measuring Social-Ecological Pathways: The Case of Global Changes and Water Security” contribute to the literature of Socio-Ecological Systems (SESs) framework, providing an institutional and cognitive model of change of SESs. The authors identify and discuss methodological limitations preventing from reliable and generalizable results of SESs. Likewise, new tools such as Earth Observation Data Cubes are proposed in order to enable the combination of social and ecological data over space and time, and overcome some limitations regarding these aspects.

Due to the complexity of the assessment of SESs and, above all, quantitative assessment, I consider the authors’ contribution as significant at this arena.

However, some aspects of the manuscript should be highly improved. Mainly, the theoretical framework should be deeply change, because a coherent discourse is missing. There are many ideas and information, but I miss, as aforementioned, a coherent discourse.

Moreover, I think authors are able to offer a more specific information about the case of global changes and water security. In other words, authors should provide more specific results about the implementation of the different approaches they explain over the case of global changes and water security. Section 4.3 may hold this improvement.

Additionally, I find that there are an excessive number of footnotes. Not all of them are essential to understand the manuscript. Please, reduce them at maximum as possible. For instance, number 8 may be removed.

Several mistakes in the references have been found, such as “[?]” (Lines 180 and 313). Please, revise the guidelines of the journal for referencing other authors.

In the following lines, I point one by one other changes that I also consider authors should be address:

1.   Introduction

-          Avoid repetition on the wording in the same sentences:

o   Lines 13-14: two times “need”

o   Lines 31-33: three times “help”

-          There is a lack of connection between the first and second paragraph.

-          Substitute “everyone” by “human” in line 17.

-          Substitute “ask” by “demand” in line 21.

-          Line 35: The sentence “SES lacks a robust understanding of change” needs further explanation and a reference.

-          Line 37: What does “large-N” means?

-          Line 38: Reformulate the sentence “…SES is needed for knowledge accumulation”. I do not consider the need is the accumulation of knowledge, but the need of this knowledge to be applied into an integral decision-making process, or critical assessment of policies and measures, for instance.

-          Lines 41-43: The paragraph from “Current literature…” to “…generalizable results” needs references.

-          Line 49: the limitations are not clear. It needs an explanation.

-          Line 51: Data Cube technology appears here by the first time in the manuscript (and in my life), and you do not mention it again until page 11. I think you need to explain briefly what it is and a reference at this point (Line 51).

2.    SES and Evolution: Theoretical Framework

This section is frankly chaotic. There are many ideas, but there is a huge lack of connection among them and a huge lack of a discourse along the whole section.

Authors need to perform a deep change in the discourse of the whole section.

Other minor comments:

-          Line 67: Authors need to make clearer the SESs framework

-          Avoid repetition on the wording in the same sentences:

o   Lines 68-69: three times “framework”-

-          You should remove the name of authors and properly mentioned the number of their reference, such as [10,11] in Line 70; [10] in Line 71.

-          Line 80: needs a reference for the first sentence.

-          Line 84: Authors adopt the 3rd generation of resilience thinking and provide a reference. However, I do not see that Folke (2006) is referred to the term generation.

-          Lines 87-88: It could be possible to eliminate the idea of generation? It is confused and, in my opinion, unnecessary.

-          Figure 1: What do the symbols (‘r’,’k’, ‘alpha’, ‘omega’) mean? If they are not important in your discourse, remove them from the figure.

-          Lines 112: explain the concept “rigidity trap”.

-          Lines 108-114: Re-write the paragraph. It is confusing.

-          Line 137: Remove the author of reference [25].

-          Line 130: I could not find the reference [25,26].

-          Lines 137-146: Include ideas of trade-offs. For example, use the discourse of Banos-González et al. (2016) and Vidal-Legaz et al. (2013).

Banos-González, I., Martínez-Fernández, J., & Esteve-Selma, M. A. (2016). Using dynamic sustainability indicators to assess environmental policy measures in Biosphere Reserves. Ecological Indicators, 67, 565-576.

Vidal-Legaz, B., Martínez-Fernández, J., Picón, A. S., & Pugnaire, F. I. (2013). Trade-offs between maintenance of ecosystem services and socio-economic development in rural mountainous communities in southern Spain: a dynamic simulation approach. Journal of Environmental Management, 131, 280-297.

-          Line 157: Table 1 needs further explanation at this point.

-          Line 165: What does “subsidiary” mean in this context?

-          Line 180: Revise reference”[?]”.

-          Line 185: Remove the authors’ names of reference [36] and re-write the sentence to avoid starting such as: “[36] recent work…”

-          Line 195: Remove the authors’ names of reference [40].

-          Figure 3: Modify the figure in order to be more attractive.

-          Line 254: Identify the context of the review performed by reference 46.

-          Line 258: Remove the authors’ names of reference [48] and re-write the sentence.

-          Line 272: Remove the authors’ names of reference [46] and re-write the sentence.

-          Line 291: Specify what does ‘N’ mean? Number of observations?

-          Line 296: Remove the authors’ names of reference [51] and re-write the sentence.

-          Line 310: Remove the authors’ names of reference [52] and re-write the sentence.

-          Line 313: Revise reference”[?]”.

-          Line 315: of an SES.

-          Line 317: Re-write the beginning “.[18] highlight…”

-          Line 318: Remove the authors’ names of reference [46] and re-write the sentence.

-          Line 326: Remove the authors’ names of reference [8] and re-write the sentence.

-          Line 317: Re-write the beginning “.[26] transpose…”

-          Line 376: the sentence between comas (“, that are increasingly… repositories,”) seems not essential. I suggest to remove it.

-          Line 378: “potential information” instead of “information potential.”

-          Line 379: “…EO data has not been yet realized” needs to be clarified or improving the connection with the next sentence.

-          Figure 4 is not essential. Remove it.

-          Footnote number 8 may be removed.

-          Figure 5: the reference should appear according to the journal’ guidelines.

4.3. Avenues for implementing an analysis of water security dynamics through Data Cube

I think this section should be one of the most important part of the work, reflecting the implementation of the different approaches that authors explain on the case of global changes and water security. However, the section misses more specific information (results) about the case of global changes and water security. It should be deeply improved.

Other minor comments:

-          Line 442: Remove the authors’ names of reference [52].

-          Line 446: Remove the authors’ names of reference [2] and re-write the sentence.

-          Line 453: Remove the authors’ names of reference [2] and re-write the sentence.

5.   Conclusion

Line 501: there is a mistake: “way consists involves”

Authors Contributions, funding, acknowledgements and conflict of interest need to be explained.

Abbreviations are incomplete. There are many other abbreviations in the manuscript which do not appear in this list.

Author Response

Response to Referee Number 2

We are very grateful for your positive view of our paper, for what you see as its three main contributions, and for the many constructive criticisms you made. We have found your comments both appropriate and extremely helpful. In the revised version of the paper, we believe that we have responded fully to your comments and suggestions. Below, we detail our responses to your specific concerns (your original comments are in italics).

This paper, entitled “Explaining and Measuring Social-Ecological Pathways: The Case of Global Changes and Water Security” contribute to the literature of Socio-Ecological Systems (SESs) framework, providing an institutional and cognitive model of change of SESs. The authors identify and discuss methodological limitations preventing from reliable and generalizable results of SESs. Likewise, new tools such as Earth Observation Data Cubes are proposed in order to enable the combination of social and ecological data over space and time, and overcome some limitations regarding these aspects.

Due to the complexity of the assessment of SESs and, above all, quantitative assessment, I consider the authors’ contribution as significant at this arena.

However, some aspects of the manuscript should be highly improved. Mainly, the theoretical framework should be deeply change, because a coherent discourse is missing. There are many ideas and information, but I miss, as aforementioned, a coherent discourse.

Thank you very much for these most useful comments. We have revised the paper extensively and accordingly. We have streamlined all the theoretical section (section 2) and punctuated it with clear statements of what we want to do and how we want to do (lines 71-81, lines 111-112, lines 122-128, figures 1 and 3).

Moreover, I think authors are able to offer a more specific information about the case of global changes and water security. In other words, authors should provide more specific results about the implementation of the different approaches they explain over the case of global changes and water security. Section 4.3 may hold this improvement.

This comment of yours proved to be extremely useful as it helped us to better formulate our contribution. We have revised the manuscript accordingly and now provide a detailed explanation of specific results about water security and the implementation of the different approaches (lines 472-539).

We have discussed in depth the water security index (Gain et al. 2016) and how data cube could contribute to expanding its scope (lines 486-501) and provided insights on crucial aspect of water management such as water quality (lines 479-480, and 506-508), water-related hazards (lines 510-521 and 526-531), and hydrological structure (lines 505-506).

We have developed and approaches implementation by underlining the role of interoperability (lines 452-457), and the ability to ‘play’ with geographical and temporal scales for the sake of setting robust empirical measures  (lines 510-521) and tests (lines 472-483, 503-508, 523-539).

Additionally, I find that there are an excessive number of footnotes. Not all of them are essential to understand the manuscript. Please, reduce them at maximum as possible. For instance, number 8 may be removed.

Several mistakes in the references have been found, such as “[?]” (Lines 180 and 313). Please, revise the guidelines of the journal for referencing other authors.

Thank you very much, we have reduced the number of footnotes from 9 to 6 and these mistakes have now been corrected.

In the following lines, I point one by one other changes that I also consider authors should be address:

Thank you very much for these most useful comments. We have revised the paper extensively and accordingly.

1.     Introduction

Avoid repetition on the wording in the same sentences:

o   Lines   13-14: two times “need”

o   Lines   31-33: three times “help”

(Line 15 and 31-33) These mistakes have now been corrected.

There is a lack of connection   between the first and second paragraph.

(Line 18)   We added a connection sentence.

Substitute “everyone” by “human”   in line 17.

(Line 17)   substituted.

Substitute “ask” by “demand” in   line 21

(Line 22)   substituted.

Line 35: The sentence “SES lacks a robust understanding of change”   needs further explanation and a reference.

(Lines   35-37) Thank you, we have clarified this

Line 37: What does “large-N”   means?

(Line 39) Thank   you, we have clarified this

Line 38: Reformulate the sentence   “…SES is needed for knowledge accumulation”. I do not consider the need is   the accumulation of knowledge, but the need of this knowledge to be applied   into an integral decision-making process, or critical assessment of policies   and measures, for instance.

(Lines   44-45) Thank you, we   have clarified this by emphasizing the need for scaling-up knowledge in that   field.

Lines 41-43: The paragraph from   “Current literature…” to “…generalizable results” needs references.

(Lines   44-45) References have been added.

Line 49: the limitations are not   clear. It needs an explanation.

(Lines51-52)   We have mentioned them: measurement heterogeneity and time-limited analysis.

Line 51: Data Cube technology appears here by the first time in the   manuscript (and in my life), and you do not mention it again until page 11. I   think you need to explain briefly what it is and a reference at this point   (Line 51).

(Lines54-57)   Thank you, we have clarified this

2.      SES and Evolution: Theoretical Framework

This section is frankly chaotic. There are many ideas, but there is a   huge lack of connection among them and a huge lack of a discourse along the   whole section.

Authors need to perform a deep change in the discourse of the whole   section.

(All section) This comment of yours proved to   be extremely useful as it helped us to better formulate our theoretical   proposition

Line 67: Authors need to make clearer the SESs framework

Avoid repetition on the wording in the same sentences:

o   Lines   68-69: three times “framework”-

(Lines 84-86) We have revised the passage   accordingly.

You should remove the name of   authors and properly mentioned the number of their reference, such as [10,11]   in Line 70; [10] in Line 71.

(Lines   84-86) We have removed the names and properly mentioned references

Line 80: needs a reference for the first sentence.

(Line   94) we added two references

Line 84: Authors adopt the 3rd   generation of resilience thinking and provide a reference. However, I do not   see that Folke (2006) is referred to the term generation.

Lines 87-88: It could be possible   to eliminate the idea of generation? It is confused and, in my opinion,   unnecessary.

(Line   97-101) Many thanks, we have modified the passage accordingly and it is   clearer.

Figure 1: What do the symbols   (‘r’,’k’, ‘alpha’, ‘omega’) mean? If they are not important in your   discourse, remove them from the figure.

We   have drawn a new figure accordingly

Lines 112: explain the concept “rigidity trap”.

Lines 108-114: Re-write the paragraph. It is confusing

(Lines 124-125) We have clarified this

Line 137: Remove the author of reference [25].

(Line   151) Removed

Line 130: I could not find the reference [25,26].

DOI   updated for [25] and [26] is available at: https://papers.ssrn.com/abstract=3163828

Lines 137-146: Include ideas of trade-offs. For example, use the   discourse of Banos-González et al. (2016) and Vidal-Legaz et al. (2013).

Banos-González,   I., Martínez-Fernández, J., & Esteve-Selma, M. A. (2016). Using dynamic   sustainability indicators to assess environmental policy measures in   Biosphere Reserves. Ecological Indicators, 67, 565-576.

Vidal-Legaz, B., Martínez-Fernández, J., Picón, A.   S., & Pugnaire, F. I. (2013). Trade-offs between maintenance of ecosystem   services and socio-economic development in rural mountainous communities in   southern Spain: a dynamic simulation approach. Journal of   Environmental Management, 131, 280-297.

(Lines 156-158) Thank you for these remarks and the suggested   readings! We have included this idea of trade-off (in lines194-196 as well)

Line 157: Table 1 needs further explanation at this point.

(Lines   173-176). Thank you, we have clarified this

Line 165: What does “subsidiary” mean in this context?

Thank   you, we have decided to delete the term because indeed there is not necessary   administrative rule that organizes the multilevel setting of learning   process, so the use of “subsidiary” created confusion.

Line 180: Revise reference”[?]”.

(Line   209) Revised

Line 185: Remove the authors’ names of reference [36] and re-write the   sentence to avoid starting such as: “[36] recent work…”

(Lines   209-212) We have modified the sentence accordingly

Line 195: Remove the authors’ names of reference [40].

(Line   221) We have removed the authors’ names

Figure 3: Modify the figure in order to be more attractive.

Figure   3. Many thanks, we have modified the figure accordingly.

Line 254: Identify the context of the review performed by reference 46

(Lines 277-278). Thank you, we have clarified this

Line 258: Remove the authors’ names of reference [48] and re-write the   sentence

(Lines   283-285) We have modified the sentence accordingly

Line 272: Remove the authors’ names of reference [46] and re-write the   sentence.

(Lines   295-297) We have modified the sentence accordingly

Line 291: Specify what does ‘N’ mean? Number of observations?

(Line 315) we have clarified this

Line 296: Remove the authors’ names of reference [51] and re-write the   sentence.

(Lines 320-322) We have modified   the sentence accordingly

Line 310: Remove the authors’ names of reference [52] and re-write the   sentence.

(Lines 333-335) We have modified   the sentence accordingly

Line 313: Revise reference”[?]”.

(Line   337) this mistake has now been correct

Line 315: of an SES.

(Line 339) this mistake has now   been correct

Line 317: Re-write the beginning “.[18] highlight…”

(Lines 341-342) We have modified   the sentence accordingly

Line 318: Remove the authors’ names of reference [46] and re-write the   sentence.

Line 326: Remove the authors’ names of reference [8] and re-write the   sentence.

(Lines 350-352) We have modified   the sentences accordingly

Line 317: Re-write the beginning “.[26] transpose…”

(Lines 353-355) We have modified   the sentences accordingly

Line 376: the sentence between comas (“, that are increasingly…   repositories,”) seems not essential. I suggest to remove it.

We have removed it

Line 378: “potential information” instead of “information potential.”

Line 379: “…EO data has not been yet realized” needs to be clarified   or improving the connection with the next sentence.

(Line 402-404) We have modified   the sentences accordingly

Figure 4 is not essential. Remove it.

Figure   4 has been removed

Footnote number 8 may be removed.

Footnote   8 has been removed

Figure 5: the reference should appear according to the journal’   guidelines

Reference   to Killough is now correct in figure 5

4.3. Avenues for implementing an analysis of water security dynamics   through Data Cube

I   think this section should be one of the most important part of the work,   reflecting the implementation of the different approaches that authors   explain on the case of global changes and water security. However, the   section misses more specific information (results) about the case of global   changes and water security. It should be deeply improved.

Thank you very much for these most useful   comments. We have revised the paper extensively and accordingly.

Line 442: Remove the authors’ names of reference [52].

Line 446: Remove the authors’ names of reference [2] and re-write the   sentence.

Line 453: Remove the authors’ names of reference [2] and re-write the   sentence

We have taken into account these 3 recommended changes, but can’t   refer to here as the section as been totally rewritten.

5.   Conclusion

Line 501: there is a mistake: “way consists involves”

(Line 583) Thank you very much,   this mistake has now been correct.

Authors Contributions, funding, acknowledgements and conflict of   interest need to be explained.

Abbreviations are incomplete. There are many other abbreviations in   the manuscript which do not appear in this list.

Thank   you, we have added these information to the current version

Round 2

Reviewer 1 Report

The authors have addressed the suggested revisions. I think that the revisions have
improved the quality of the current manuscript and I do not have any additional
comment.

Reviewer 2 Report

I think the athors have adequatly addressed the changes suggested by this Reviewer. Therefore, I accept the manuscript in the present form.